# The spatial heterogeneity of urban green space distribution and configuration in Lilongwe City, Malawi

**Odala Nambazo**[1]*, **Kennedy Nazombe**[2]

**1** Department of Land Resources Conservation, Ministry of Agriculture, Lilongwe, Malawi, **2** Faculty of Life Sciences and Natural Resources, Department of Land and Water Resources, Natural Resources College, Lilongwe University of Agriculture and Natural Resources, Lilongwe, Malawi

* nambazo@yahoo.com

**Data Availability Statement:** The data can be accessed on a public repository via the following link: https://data.mendeley.com/datasets/rfkv3y4237/1.

## Abstract

Urban green spaces provide several benefits related to the quality of urban life. The existence and spatial arrangement of these spaces within neighbourhoods and functional land uses have significant implications for the well-being of urban dwellers. Previous studies on green spaces in urban areas of Malawi have focused on a broader and macro-level perspective, offering insightful information on general trends in different cities. However, there is a significant research shortage in localised understanding, which requires carrying out micro-level assessments concentrating on land use zones and neighbourhoods within these cities. In this study, we used remote sensing data and landscape metrics to understand the distribution and configuration of urban green spaces in the city's neighbourhoods and functional land uses and their relationship with urban form. The study revealed that 20% of neighbourhoods fail to meet the WHO-recommended standard of 9 m² of green space per person, with a predominant concentration of these undersupplied areas in high-density and quasi-residential zones. In addition, 56.2% of Lilongwe City's total green area was contained under functional land uses. Particularly, high-rise residential, medium-density residential, low-density residential, quasi-residential, high-rise flat area, commercial class, high-rise commercial, heavy industry, light industry, and government land use zones contained 17.3%, 12.0%, 22.2%, 12.0%, 4.1%, 6.4%, 6.1%, 5.0%, 1.6%, and 13.3% of the total green spaces in functional land uses, respectively. Importantly, this research found significant correlations between urban form metrics, namely building coverage, building density, building perimeter area ratio, road density, and the distribution and configuration of urban green spaces. This necessitates an integrated approach to urban planning and design, emphasising the importance of balancing development with green space preservation.

## Introduction

Urbanisation is the key characteristic of the twenty-first century, as people migrate to cities worldwide in search of better living conditions and employment opportunities [1]. However,

**Funding:** The author(s) received no specific funding for this work.

**Competing interests:** The authors have declared that no competing interests exist.

there are several consequences of rapid urbanisation, including increased environmental degradation, increased socioeconomic disparity, and the loss of natural areas that are essential to the well-being of urban residents [1,2]. The design and distribution of urban green spaces (UGS) become increasingly important factors that impact the quality of life as cities expand and evolve [3,4].

The equitable distribution and use of UGS are becoming more important for the sustainability of cities. However, UGS are not always evenly and equitably distributed among city dwellers [5]. Urban green spaces can be anything from parks and gardens to greenways. Urban green spaces offer multiple benefits to city dwellers. These include; improving air quality [6], providing recreational opportunities [7], fostering ecological balance in densely populated areas [8], enhancing the aesthetic appeal [9], providing educational opportunities [10], and enhancing psychological well-being [11]. To harness the benefits of UGS, urban planners are now interested in harmonious coexistence between the natural and built environments. Healthy and happy city dwellers are the fundamental goals of city design [12].

Numerous studies have linked the variability of UGS across different land use zones and neighbourhoods to several factors. For instance, urbanisation, population density, and land value play pivotal roles in the availability of UGS in various locations [13–15]. In addition, environmental factors, governance, regulatory policies, and zoning restrictions affect the distribution of green areas within different zones [16–18]. Socioeconomic factors, such as race and income disparities, also influence the availability and quality of UGSs [19–22]. Since the distribution of green space in urban areas tends to be uneven due to numerous factors, mapping the current state of green space is necessary for urban planners to create sustainable cities.

Previous research on green space and land use/land cover in cities and towns has mostly provided a macro-level perspective in Malawi [23–26]. These studies offered insightful information on general trends in different cities and towns. However, localised assessments within Lilongwe´s diverse neighbourhoods and functional land uses remain scarce, hindering targeted urban planning initiatives. Both formal and informal settlements with a varied range of densities and distinct socioeconomic diversity among their inhabitants characterise the urban fabric of Lilongwe City [27]. Compared to cities in developed regions, these distinct attributes might have a very different effect on the supply and demand for UGS. Such a localised approach is essential because it recognises that the dynamics of green spaces can differ significantly within a city's boundaries, depending on a variety of variables such as historical development, changing land-use regulations, community-driven initiatives, and social-economic factors [13,18,19,21,28]. This knowledge gap underscores the need for studies examining the specific settings of neighbourhoods or communities, providing a deeper understanding of how green spaces are distributed in different areas of the city.

The present study investigates an aspect of urban green space management and planning in Lilongwe City. Our focus was on the composition and configuration of UGSs in the city's neighbourhoods and functional land uses, and how the urban form affects these factors. According to the city´s urban structure plan, the following functional land uses were included in the study: commercial, industrial, government, and residential areas. The residential areas were further classified into high-density, medium-density, low-density, quasi-residential, and high-rise residential areas. The objectives of the study were to analyse (1) the distribution of urban green spaces in the city's neighbourhoods and functional land uses; (2) the configuration of urban green spaces in functional land uses; and (3) the relationship between urban form and green space distribution and configuration. We used remote sensing and Geographic Information Systems (GIS) techniques to map the distribution and configuration of UGS. Remote sensing and GIS have proven to be efficient and effective means to characterise UGS

in terms of abundance, spatial distribution, and species composition [26,29–32]. Geospatial techniques facilitate precise mapping and assessment of spatial patterns and configurations of UGS [26,32]. The findings of this study may have an impact on urban planning strategies and policies and aid in the creation of more efficient plans for the distribution and management of UGS.

## Materials and methods

### Study area

Lilongwe was declared the capital city of Malawi in 1975, after relocating from Zomba City. The city lies between 13°45′S and 14°3′ S latitude, 33°41′E and 33°53′ E longitude in the central region of Malawi (Fig 1). It is the largest city in Malawi with a population of 989,318 [33] and covers approximately 727.8 km$^2$. The city was designed based on the garden city concept and has abundant greeneries within the central part of the city [27]. The Japan International Cooperation Agency (JICA) in conjunction with the Malawi Government prepared the Urban Development Master Plan in 2010 to guide the city's development. According to the city's

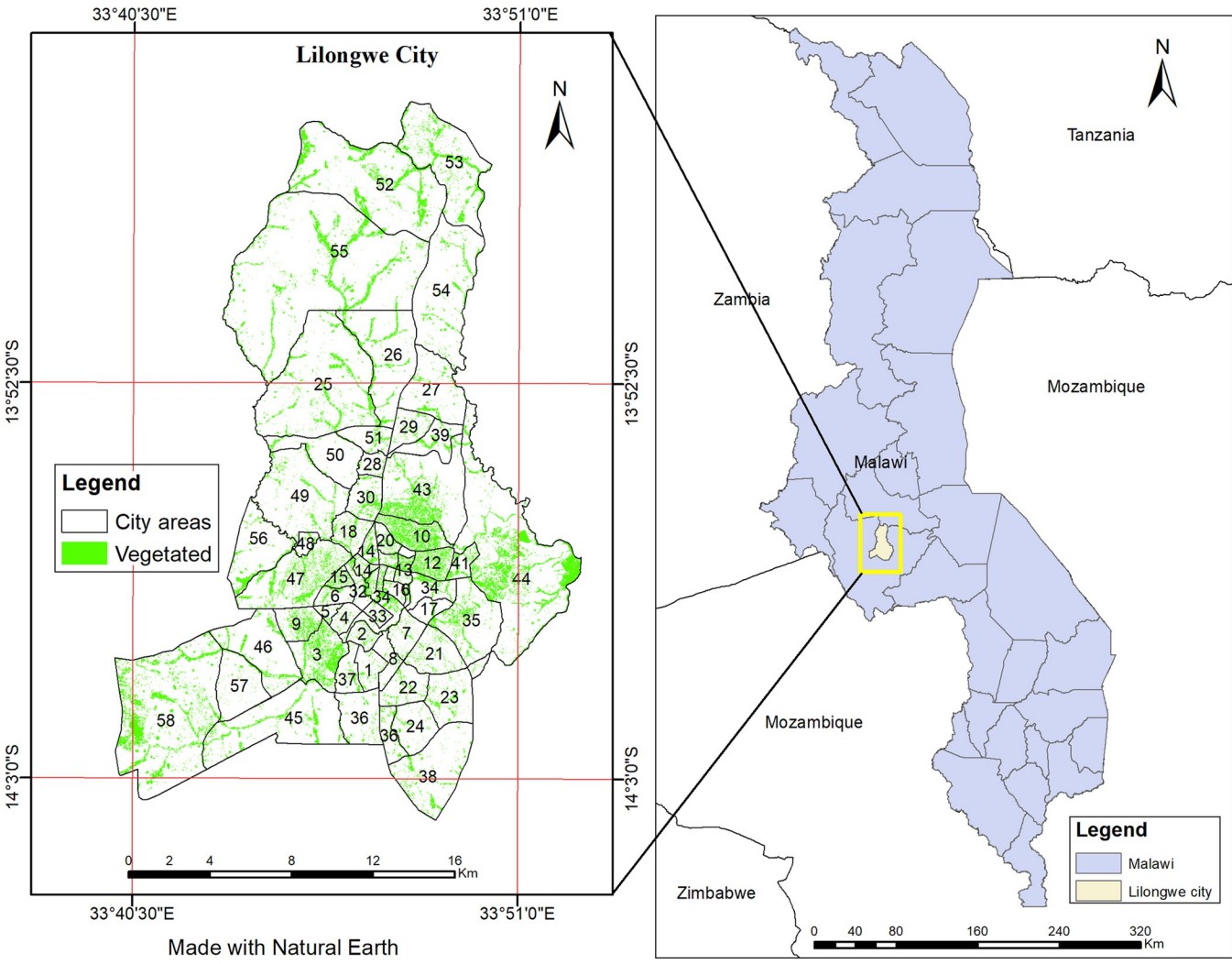

**Fig 1. Location and areas of Lilongwe city (Malawi) and the green spaces within it.**

plan, about 22,000 ha of land was zoned for nature sanctuary, parks and recreation, greeneries, agriculture, and forestry land uses. The city is abounding in natural beauty and environmental resources, including rivers, streams, flowers, and natural trees. However, the city is experiencing rapid urbanisation compared to other cities in Malawi, which is estimated at 4% annually [34]. Like many other cities, Lilongwe is experiencing many challenges including those caused by rapid urbanisation. Environmental deterioration, pollution, deforestation, uncontrolled development in ecologically sensitive, and weak regulatory frameworks are among the other challenges facing the city.

## Data

We used a 10-metre-resolution Sentinel Level-2A imagery acquired on September 5, 2022 (Product ID: L2A_T36LXK_A028718_20220905T080208). A satellite image covering the entire city of Lilongwe was obtained from the Copernicus Open Access Hub (https://scihub.copernicus.eu/). To minimise haze and acquire cloud-free satellite images of the study area, the image was acquired during the dry season. The image was then projected onto the World Geodetic System 1984, the Universal Transverse Mercator (UTM) Zone 36S. The Sentinel Level-2A products are already subjected to atmospheric and geometric corrections. The acquired satellite images were clipped to extract study areas using the Lilongwe city boundary shapefile layer obtained from the Lilongwe City Council. We used Microsoft building footprint data, which were downloaded from https://www.microsoft.com/en-us/maps/building-footprints. The city's neighbourhood's boundary shapefile was obtained from the National Statistical Office (NSO), which was based on census tract data. There is no universally agreed-upon definition of a neighbourhood and most studies have used census boundaries [35]. The population census data for the neighbourhoods were extracted from the 2018 Malawi Population and Housing Census Report [33]. This study adapted the four main functional land use categories outlined in the city's urban structure plan [27] (Table 1). The general methodology is outlined in Fig 2.

## Methods

**Normalised difference vegetation index.** We used the normalised difference vegetation index (NDVI) to identify green spaces in the study area. NDVI is one of the most used indicators of the presence of vegetation [22,29–32]. The NDVI has a range of -1 to +1. A negative NDVI indicates non-green areas, such as deserts, water, rivers, and built-up areas, whereas a positive value denotes green areas [36] and increased NDVI value suggests more vegetation on the ground. The NDVI can also be used to assess the conditions of plants and vegetation [37].

**Table 1. Descriptions of the functional land use categories used in the study.**

| Functional land use categories | Description |
| --- | --- |
| Residential (High-Density Residential, Medium Density Residential, Low-Density Residential, Quasi Residential, and High-Rise Flat Area) | Land primarily dedicated to housing, such as single-family homes, apartment complexes, and residential neighbourhoods. |
| Commercial | It encompasses areas where businesses and retail activities take place, such as small shops, shopping malls, restaurants, banks, and office buildings. |
| Industrial (Heavy/Large Scale Industry and Light Industry) | Areas where manufacturing, warehousing, and other industrial activities occur. |
| Government | Government land use includes government office buildings where various government functions and public services take place and includes state residences. |

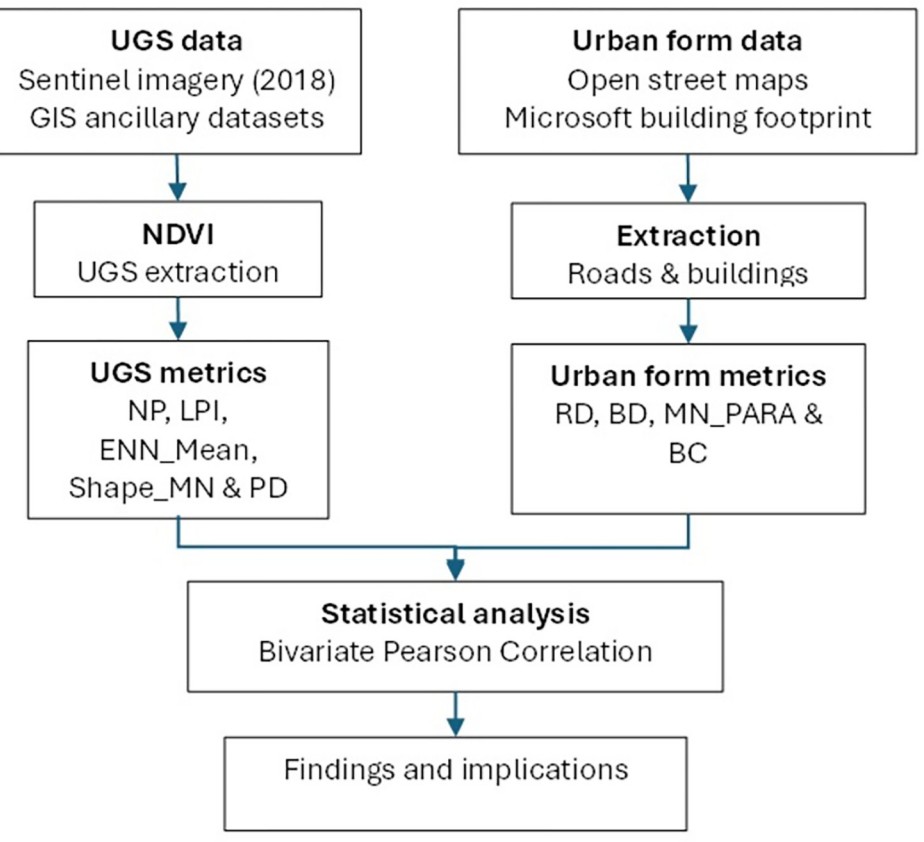

**Fig 2. Methodological framework applied for this study.**

This is because healthy plants have greater NDVI values than stressed plants. Eq (1) was used to calculate the NDVI in ArcGIS 10.6 software as follows:

$$\text{NDVI} = \frac{\text{NIR[Band 8]} - \text{RED[Band 4]}}{\text{NIR[Band 8]} + \text{RED[Band 4]}} \tag{1}$$

where,

NIR [Band 8] and RED [Band 4] denote the reflectance in the near-infrared and red bands, respectively, of the Sentinel-2A imaging product.

**Per Capita Green Space (PCGS) and Urban Green Space Index (UGSI).** To understand green space allocation for every inhabitant of a neighbourhood, we calculated per capita green space (PCGS). The PCGS is widely used to assess the quality of urban environments and their impact on residents' well-being [13,26,32,38,39]. Eq (2) was used to calculate the PCGS:

$$PCGS_i = \frac{G_i}{PN_i} \tag{2}$$

where $PCGS_i$ denotes per capita green space in neighbourhood $i$, $G_i$ denotes total green space coverage in neighbourhood $i$, and $PN_i$ denotes the total population of neighbourhood $i$. We used population census data for neighbourhoods for the year 2018 [33].

Furthermore, the Urban Green Space Index (UGSI) was adopted to measure the quantity of Urban Green Spaces (UGS) in each neighbourhood. The index denotes the availability of UGS

as a percentage and provides a standardised way of comparing UGS in different neighbourhoods [32]. The UGSI is computed as follows:

The UGSI for $i^{th}$ neighbourhood can be expressed as in the Eq (3):

$$UGSI_i = \frac{G_i}{A_i} \qquad (3)$$

Where, $G_i$ = UGS in the neighbourhood $i$, $A_i$ = area of the $i^{th}$ neighbourhood (where $i$ = 1 to n).

The total UGS (expressed as a percentage) in a particular neighbourhood is calculated as in the Eq (4):

$$UGSI_T = \frac{\sum_t^n G_i}{\sum_t^n A_i} \times 100 \qquad (4)$$

**Urban green space landscape metrics.** Guided by previous studies [5,40,41], the composition and configuration of urban green spaces were assessed using the following seven landscape metrics: number of patches, largest patch index (LPI), mean Euclidean Nearest Neighbour distance (ENN_MEAN), mean shape index (SHAPE_MN), number of patches (NP), and patch density (PD). We used FRAGSTATS 4.2 software to calculate the selected metrics of green spaces at the class level. A detailed description of the metrics is provided in Table 2.

**Urban form metrics.** Several studies have shown significant impacts of urban form on the spatial distribution and configuration of UGS. The relationship between urban form and green spaces has implications for the overall sustainability, liveability, and well-being of a city. Weighted density, density gradient slope, density gradient intercept, compactness, and street connectivity urban form metrics were used to understand the impact of urban form on green space accessibility in 462 metropolitan areas globally [42]. To understand the associations between urban morphology and green spaces, building coverage ratio, building perimeter, the number of buildings, road coverage ratio, road intersections, and road length ratio metrics were employed [43]. Similarly, the urban form indicators of address density, building density, and household density were correlated with the biodiversity potential and ecosystem performance indicators in five UK cities [44]. The perimeter-area ratio (PARA), road density (RD), and compound terrain complexity index were used to evaluate the impact of urban form on the UGS structure [45]. In Sheffield, road length, building density, and building area, among other factors, are predictors of the extent and quality of green spaces [29]. Therefore, four

**Table 2. A detailed description of the landscape metrics.**

| Category | Landscape Indices | Description |
|---|---|---|
| Composition | Number of Patches (NP) | Quantifies the number of green patches existing in a particular study unit. The higher NP suggests more fragmentation. |
| | Largest Patch Index (LPI) | The proportion of the largest green patch with a study unit. It indicates the level of dominance and concentration in a study unit. |
| Configuration | Mean Euclidean nearest neighbour distance (ENN_MEAN) | It measures the average distance from one green space patch to the nearest green space patch. It indicates the level of dispersion or clustering of green spaces. |
| | Mean shape Index (SHAPE_MN) | The average shape complexity of green space patches in a study unit. More irregular shapes are associated with higher shape index values. |
| | Patch Density (PD) | It quantifies the total number of green space patches distributed across an area. It shows the extent of spatial distribution, connectivity, and fragmentation. |

**Table 3. A detailed description of the urban form metrics used in the study.**

| Metric | Description | Relevance |
|---|---|---|
| Building Coverage (BC) | The proportion of the ground area covered by buildings to the total area of the plot and usually expressed as a percentage. | Provides an idea of the compactness and intensity of development. |
| Road Network Density (RD) | The total length of roads in an area divided by the total land area of that area. It is typically expressed in kilometres per square kilometre. | Gives insights into an area's accessibility and connectivity. |
| Mean Perimeter Area Ratio (PARA_MN) of Buildings | The average ratio of the perimeter of buildings to their area. | Highlights the complexity and configuration of building shapes. |
| Building Density (BD) | The number of buildings per unit area. Usually measured in buildings per hectare or square kilometre. | Measures the intensity of built-up development. A higher building density can indicate a more compact urban form, possibly leading to more efficient land use. |

urban form metrics were employed in the study, namely: building coverage (BC), road network density (RD), the mean perimeter area ratio (PARA_MN) of buildings, and building density (BD) (Table 3). The calculation of the urban form metrics was performed in ArcGIS 10.6 software.

Urban planners and designers use building coverage (BC) as an indicator to quantify the total area of land occupied by buildings. The amount of land covered by buildings provides information about how intensively an area is developed and how much of the land area is undeveloped. Lower building coverage enhances the preservation of open spaces, parks, and natural areas. Using Microsoft building footprint data, BC was calculated using the following Eq (5):

$$BC = \frac{BA}{TA} \tag{5}$$

where BC, BA, and TA are building coverage, the total area covered by all buildings, and the total area occupied by a particular functional land use category, respectively.

The road network density (RD), which is the ratio of an area's total road network's length to its land area, was determined using freely available road network data accessed from the OpenStreetMap (OSM) website (www.openstreetmap.org). The RD is calculated as in Eq (6).

$$RD = \frac{L}{A} \tag{6}$$

where RD is the road density, L is the total length of roads in a particular land use, and A is the total land area of a particular land use.

The mean perimeter-area ratio (PARA_MN) of buildings is one of the urban form indicators used to understand the efficiency and compactness of urban development. A lower PARA_MN value indicates the compactness of buildings in an area. Compact urban forms are frequently linked to effective land use, lower infrastructure costs, and increased walkability. PARA for each building is calculated using the Eq (7):

$$PARA_{Building} = \frac{P_{building}}{A} \tag{7}$$

Where PARA is the perimeter-area ratio of each building, $P_{building}$ is the perimeter of each building, and $A_{building}$ is the total area covered by the building. The mean perimeter-area ratio

(PARA_MN) of buildings within a particular functional land use area was computed using the following Eq (8):

$$PARA\_MN_{Buildings} = \frac{\sum_{i=1}^{n} PARA_{Building_i}}{n} \tag{8}$$

where $n$ is the total number of buildings in a particular functional land use category.

In addition, building density (BD), which is used to measure the concentration of buildings in a particular area was adopted as an indicator of urban form. BD has an impact on the distribution and composition of urban greenery. Land for green spaces may be scarce in high-density areas, while low-density areas may provide space for greenery. However, low-density environments may encourage urban sprawl. BD is calculated using the Eq (9) below. It is then expressed as the number of buildings per given area:

$$BD = \frac{NB}{A} \tag{9}$$

where BD is the building density, NB is the number of buildings in each area, and A is the total land area.

**Statistical analysis.** To understand an association between urban green space metrics and urban form metrics, bivariate Pearson correlation analysis was performed using IBM SPSS 20.0, and the relevant tables were prepared. The bivariate Pearson correlation indicates the statistical significance, degree, and direction (increasing or decreasing) of a linear relationship between two continuous variables. A correlation coefficient of 1 indicates a positive correlation between the variables; a coefficient of -1 indicates a negative correlation; and a correlation coefficient of 0 indicates no association [46].

## Results

### The per capita distribution of urban green space in neighbourhoods in Lilongwe city

The analysis of UGS per capita distribution showed significant variations in the availability of green space in the Lilongwe City neighbourhoods (Table 4). The study found that 20% of the neighbourhoods have UGS which is less than the World Health Organisation (WHO) requirement of 9 m$^2$ per person. The majority of the neighbourhoods (54.5%) are provided with over 100 m$^2$ of green space per inhabitant. The remaining 25.5% of neighbourhoods had between 9 and 100 m$^2$ of green space per person, falling between these two extremes.

**Table 4. Urban green space per capita distribution in the neighbourhoods/areas in Lilongwe City (Note: Two areas were not included due to the lack of population data).**

| Per Capita UGS (person/m$^2$) | Area Numbers (arranged in ascending order based on the PCGS) | No of Areas | % of Areas | Remarks |
|---|---|---|---|---|
| Less than 9 | 57, 33, 50, 28, 36, 8, 24, 7, 23, 21 and 22 | 11 | 20.0 | 9 m$^2$/person is the WHO standard |
| 9–20 | 56, 51, 1 and 25 | 4 | 7.3 | |
| 21–40 | 48, 49 and 38 | 3 | 5.5 | |
| 41–60 | 18, 31 and 39 | 3 | 5.5 | |
| 61–100 | 53, 29, 39 and 44 | 4 | 7.3 | |
| Above 100 | 2, 4, 5, 6, 46, 54, 58, 27, 26, 47, 55, 30, 35, 41, 17, 15, 45, 9, 43, 52, 3, 37, 10, 20, 12, 34, 11, 13, 14, and 32 | 30 | 54.5 | |
| Totals | | 55 | 100.0 | |

## Urban green space index (UGSI) of different areas/neighbourhoods in Lilongwe City

A further analysis of the distribution of UGS in the neighbourhoods indicated different percentages of coverage (Table 5 and Fig 3). The results show that 43.9% of neighbourhoods have less than 10% of their area covered by green space, 33.3% have between 10% and 20%, 14.0% have between 21% and 30%, and 8.8% have between 30% and 33%.

## The relationship between population density and urban green space

We used Pearson's correlation analysis to determine the relationship between population density and the indicators and metrics of urban green spaces (Table 6 and Fig 4). The study revealed that the population density was weakly negatively correlated with the number of patches (-0.286*). This implies that areas with high population density have a low number of patches of green space. Population density was also significantly correlated with PCUGS and UGSI, with correlation coefficients of -0.367** and -0.383**, respectively. This implies that an increase in population density results in a reduction in urban green spaces. Additionally, there was a moderate negative correlation that was significant (-0.398**) between population density and mean shape index.

## Distribution of urban green space in functional land uses in Lilongwe city

The study has shown variations in the distribution of urban green spaces in functional land uses in Lilongwe City (Table 7). The study revealed that 59.7% of Lilongwe City's total green area was contained under functional land uses. In contrast to other functional land uses, the largest proportion of green spaces were found in residential neighbourhoods. In addition, 67.6% of UGS were found in residential zones. Particularly, 17.3%, 12.0%, 22.2%, 12.0%, and 4.1% of UGS were found in high-rise flat areas, medium-density residential areas, low-density residential areas, quasi-residential areas, and low-density residential zones, respectively. Similarly, green spaces covered 12.5%, 6.6%, and 13.3% of the area in commercial, industrial, and government land uses, respectively.

## Landscape metrics analysis for the UGS for four land use classes

The spatial patterns, compositions, and configurations of the nine land use classes varied according to the results of the landscape metric (Table 8). The number of green space patches (NP) found within each functional land use area varied significantly, with the highest number of patches recorded in the residential land use zones (10,226 patches). This indicates that NP in residential areas made up 71.6% of all patches in the study area, with high-density residential areas having the most patches and low-density residential areas having the fewest patches. For the PD, high-rise commercial areas had the greatest PD (133.8), followed by medium-

**Table 5.** *Distribution of UGSI in Lilongwe city at the area/neighbourhood level.*

| Urban Green Spaces | Area Numbers (arranged in ascending order based on the percentage of green space) | No of Areas | % of Areas |
|---|---|---|---|
| Less than 10% | 57,54, 50, 28, 8, 24, 38, 27, 1, 23, 7, 21, 36, 55, 49, 48, 22, 25, 45, 26, 2, 4, 29, 39, 33 | 25 | 43.9 |
| 10–20% | 56, 46, 19, 51, 16, 39, 5, 6, 35, 41, 17, 52,53, 18,58,30, 43, 44, 47 | 19 | 33.3 |
| 21–30% | 9, 31, 3, 37, 12, 34, 11 and 13 | 8 | 14.0 |
| 31–43% | 15, 14, 32, 10 and 20 | 5 | 8.8 |
| Totals | | 57 | 100.0 |

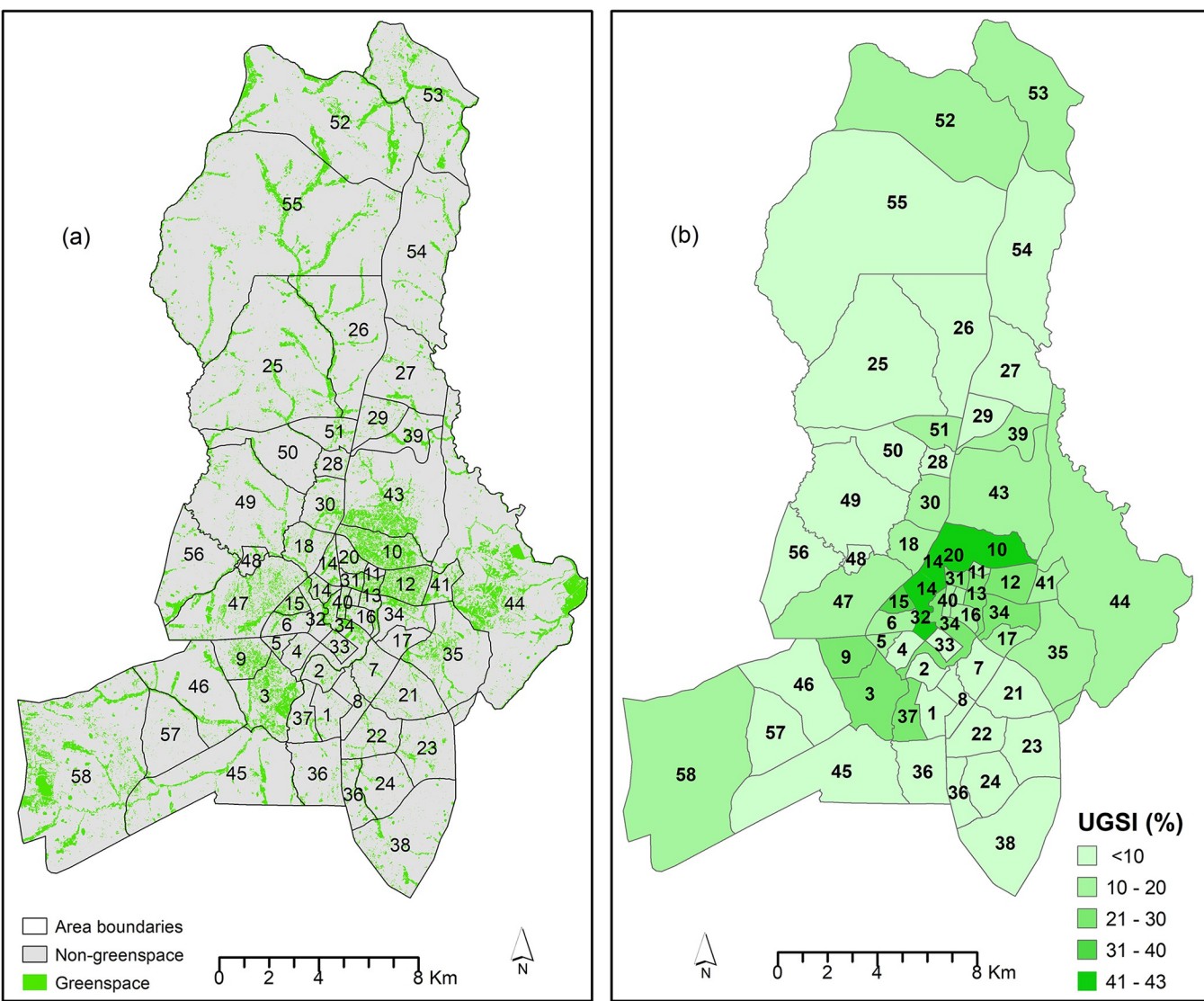

**Fig 3. Urban green space distribution.** (a) Urban Green Space at area level (NDVI derived from publicly available Sentinel-2 images) (b) Urban Green Space Index expressed in percentage at the area level.

density residential areas (PD of 110.3), and high-rise flat areas had the lowest PD (43.6). In terms of the largest patch index (LPI), low-density residential land use areas had the highest index of 9.81 and the quasi-residential class was the land use category with the smallest patch

**Table 6. *Pearson correlation between population density and urban green space metrics.***

| Variables | | Population density | NP | PD | PCUGS | UGSI | LPI | SHAPE_MN | ENN_MN |
|---|---|---|---|---|---|---|---|---|---|
| Population density | Pearson Correlation | 1 | -.286* | -.069 | -.367** | -.383** | -.235 | -.398** | .147 |
| | Sig. (1-tailed) | | .033 | .332 | .008 | .006 | .067 | .005 | .176 |
| | N | 55 | 55 | 55 | 55 | 55 | 55 | 55 | 55 |

*. Correlation is significant at the 0.05 level (1-tailed).

**. Correlation is significant at the 0.01 level (1-tailed).

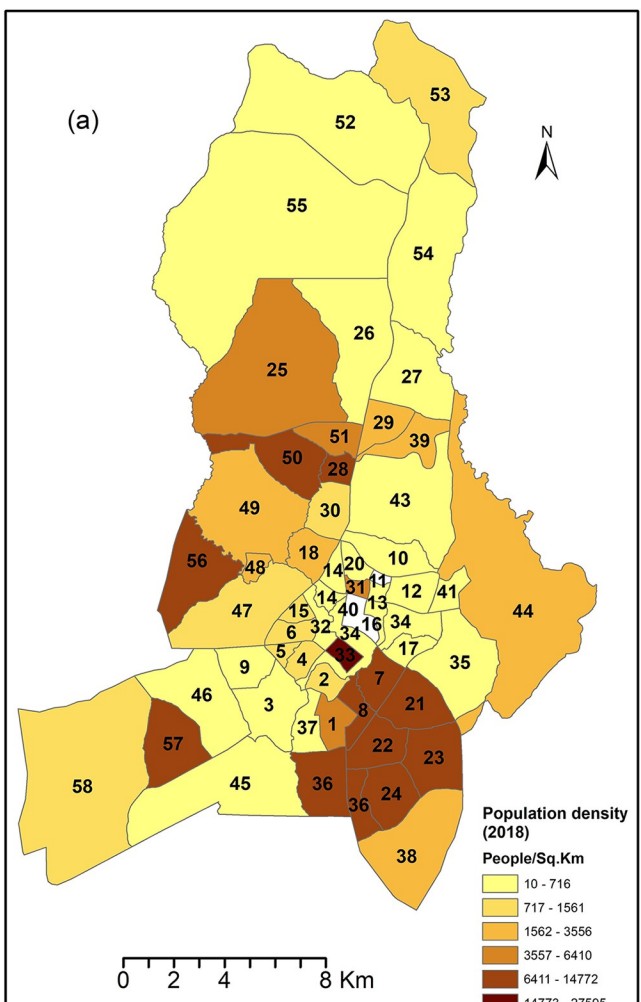
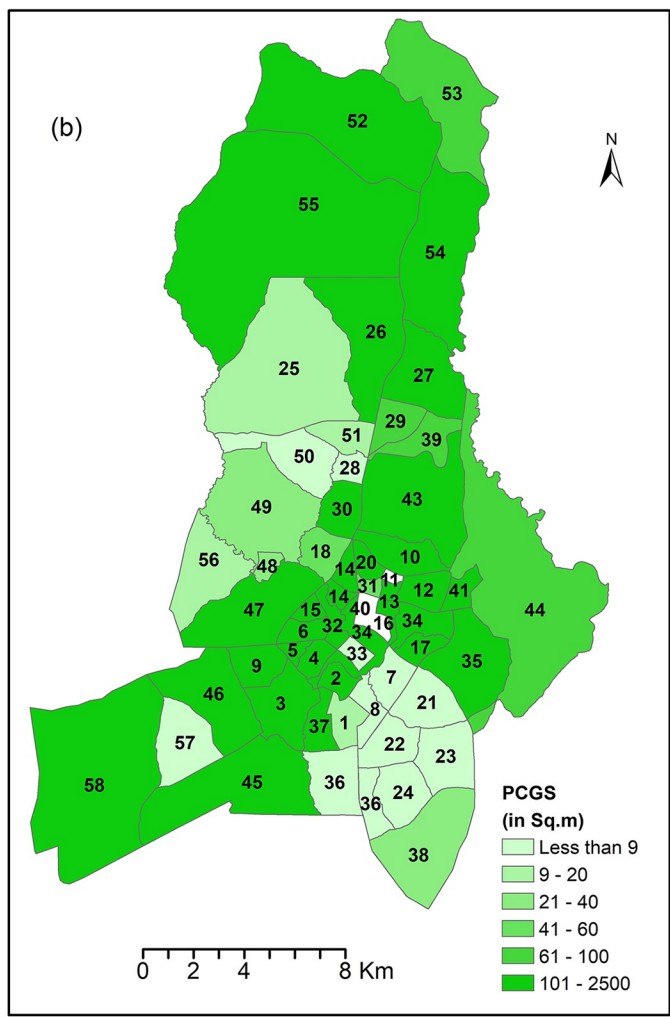

**Fig 4. Relationship between population density and PCGS.** (a) Population density map of Lilongwe City prepared from the Malawi population and housing census report [26] by the authors, (b) Per Capita Green Space distribution of Lilongwe City.

size, with a value of 0.35. A higher LPI rating for low-density residential areas denotes the presence of larger and better-connected green spaces in the landscape. On the other hand, lower LPI values imply that the green space is more fragmented, with smaller patches dispersed over the landscape. A higher LPI rating for low-density residential denotes the presence of larger and better-connected green spaces in the landscape.

In addition, the mean shape index (SHAPE_MN), an indicator of spatial configuration, showed the high complexity of the shapes of urban green spaces in all the functional land uses. The low-density residential areas had greenery that was more complex and irregular in shape (SHAPE_MN = 1.32) than the other functional land use categories. In contrast, the lowest green space shape complexities were observed in the quasi-residential areas (SHAPE_MN = 1.17).

The analysis of the mean Euclidean nearest neighbour distance (ENN_MN) indicated that quasi-residential areas had the highest value of 45.54; similarly, quasi-residential and high-rise flat residential areas had high values among the other functional land use types in the study area. This indicates that green spaces in quasi-residential areas are more dispersed or farther

**Table 7.** *Distribution of urban green spaces in functional land uses in Lilongwe City.*

| | Functional Land Use Types | Land Area (ha) | Land (%) | UGS Area (ha) | UGS Area (%) |
|---|---|---|---|---|---|
| 1 | **Residential Class** | | | | |
| | 1.1 High Density Residential | 6608.27 | 28.0 | 490.99 | 17.3 |
| | 1.2 Medium Density Residential | 1892.32 | 8.0 | 340.01 | 12.0 |
| | 1.3 Low Density Residential | 2651.61 | 11.2 | 627.66 | 22.2 |
| | 1.4 Quasi Residential | 6026.44 | 25.5 | 339.51 | 12.0 |
| | 1.5 High Rise Flat Area | 1145.82 | 4.9 | 117.36 | 4.1 |
| 2 | **Commercial Class** | | | | |
| | 2.1 Commercial Area | 1600.44 | 6.8 | 180.48 | 6.4 |
| | 2.2 High Rise Commercial | 214.58 | 0.9 | 173.84 | 6.1 |
| 3 | **Industrial Class** | | | | |
| | 3.1 Heavy/Large Scale Industry | 1324.83 | 5.6 | 141.45 | 5.0 |
| | 3.2 Light Industry | 427.00 | 1.8 | 45.07 | 1.6 |
| 4 | **Government Class** | 1733.35 | 7.3 | 375.41 | 13.3 |
| | Totals | 23624.66 | 100.0 | 2831.78 | 100.0 |

apart than those in areas with other land uses. The high value of ENN_MN indicates low clustering or concentration of green areas in the quasi-residential land use zones. On the other hand, low-density residential areas had the lowest ENN_MN value of 26.53.

## Relationship between landscape metrics of urban green spaces and urban form metrics

The UGS metrics showed some significant relationships with the urban form metrics (Table 9). For instance, RD was strongly negatively correlated with SHAPE_MN (-0.663*). BD was strongly positively correlated with NP (0.714*), SHAPE_MN (0.681*) and ENN_MN (0.651*). This suggests that as building density increases, the number of patches (NP) in green spaces tends to increase as well. In addition, in areas with higher building density, the green spaces are farther apart from each other on average. MN_PARA had a strong negative correlation with PD (-0.785**), implying that areas with buildings that have a higher mean perimeter area ratio (indicating more irregularly shaped buildings or less open space) have lower patch density in terms of green spaces.

**Table 8.** *Composition and configuration of UGS in functional land use in Lilongwe City.*

| | Land Use Types | NP | PD | LPI | SHAPE_MN | ENN_MN |
|---|---|---|---|---|---|---|
| 1 | **Residential Class** | | | | | |
| | 1.1 High Density Residential | 3584 | 63.9 | 0.51 | 1.17 | 41.5 |
| | 1.2 Medium Density Residential | 1976 | 110.3 | 2.30 | 1.25 | 28.3 |
| | 1.3 Low Density Residential | 1387 | 82.5 | 9.81 | 1.32 | 26.5 |
| | 1.4 Quasi Residential | 2779 | 55.3 | 0.35 | 1.17 | 45.5 |
| | 1.5 High Rise Flat Area | 500 | 43.6 | 1.60 | 1.26 | 41.3 |
| 2 | **Commercial Class** | | | | | |
| | 2.1 Commercial Area | 1301 | 89.3 | 0.71 | 1.20 | 35.0 |
| | 2.2 High Rise Commercial | 287 | 133.8 | 3.99 | 1.25 | 28.54 |
| 3 | **Industrial Class** | | | | | |
| | 3.1 Heavy/Large Scale Industry | 679 | 51.3 | 1.31 | 1.21 | 39.92 |
| | 3.2 Light Industry | 217 | 50.8 | 2.04 | 1.22 | 35.73 |
| 4 | **Government Class** | 1568 | 90.0 | 4.40 | 1.25 | 31.22 |

**Table 9. Correlations between urban green space and urban form metrics.**

|   |          | 1       | 2        | 3       | 4       | 5      | 6     | 7      | 8     | 9 |
|---|----------|---------|----------|---------|---------|--------|-------|--------|-------|---|
| 1 | NP       | 1       |          |         |         |        |       |        |       |   |
| 2 | PD       | -.076   | 1        |         |         |        |       |        |       |   |
| 3 | LPI      | -.230   | .353     | 1       |         |        |       |        |       |   |
| 4 | SHAPE_MN | -.484   | .335     | .874**  | 1       |        |       |        |       |   |
| 5 | ENN_MN   | .326    | -.779**  | -.744*  | -.754*  | 1      |       |        |       |   |
| 6 | RD       | .269    | .189     | -.445   | -.663*  | .276   | 1     |        |       |   |
| 7 | BD       | .714*   | -.336    | -.424   | -.681*  | .651*  | .471  | 1      |       |   |
| 8 | MN_PARA  | .111    | -.785**  | -.260   | -.207   | .480   | -.347 | .097   | 1     |   |
| 9 | BC       | .410    | .113     | -.335   | -.557   | .309   | .435  | .734*  | -.322 | 1 |

*. Correlation is significant at the 0.05 level (2-tailed).

**. Correlation is significant at the 0.01 level (2-tailed).

## Discussions

### Urban green space distribution in Lilongwe City

Lilongwe City is one of the cities with abundant greenery. Despite having abundant greenery, there are disparities in the distribution of greenery among neighbourhoods and functional land uses. The study revealed that 20% of the neighbourhoods did not meet the minimum WHO-recommended value of 9 m$^2$ of green space per individual [47]. This means that the available green spaces contribute little to the positive living in these neighbourhoods. It is worth noting that most of these neighbourhoods are in high-density and quasi-residential areas. These neighbourhoods contain approximately 50% of the total population of the city [48]. This shows how serious the disparities are in the distribution of UGS in the city. Unequal accessibility to urban green space among urban dwellers is recognised as an environmental justice issue [49].

In addition, the study revealed that population density was negatively correlated with urban green space. Numerous factors, such as a shortage of adequate land in densely populated neighbourhoods or encroachment on areas allocated for greenery, may contribute to this [27]. The quasi-residential areas, which are informal settlements, and some of these areas have encroached on environmentally sensitive areas with little space to provide them with green space. These results are consistent with those of Bille et al. [13], who investigated how population density affects patterns in urban green space around the world. They found a rapid decline in UGS coverage in areas with high population densities. Other authors have also observed the opposite effect of population on urban green spaces [50,51].

In Lilongwe City, most of the people who live in these undersupplied areas (quasi and high-density residential neighbourhoods) are low-income residents. In Lilongwe City, 73% of the residential land is composed of informal settlements and low-income neighbourhoods [48]. Disparities in green space distributions were also found in Southeast Asia [52], South Africa [21], Guangzhou, China [22], and Brisbane, Adelaide, Sydney, Perth, and Melbourne [53], with low availability of greenery found in low-income neighbourhoods. Similarly, the poorest neighbourhoods in Hong Kong were observed to have limited access to green and blue spaces [54].

Urban dwellers who are deprived of the many benefits of UGS may be concerned about uneven distribution and a lack of UGS. The results highlight the challenges and need for sustainable urban planning and development to ensure equity in the distribution and accessibility of UGS to all city residents. Lack of access to green spaces in these densely populated areas can exacerbate already-existing health and environmental injustice disparities.

## Urban green composition and configuration

Regarding the composition of UGS in functional land uses, the study has revealed that approximately 59.7% of UGS in Lilongwe City are found in functional land uses. Of this, 67.6% of UGS were found in residential zones. A high proportion of UGS in the residential areas was due to their relatively large areas. In functional land uses, residential usage accounts for about 77% of the total area.

In addition, the mean shape index (SHAPE_MN), which is one of the indicators of spatial configuration, showed the high complexity in the shapes of urban green spaces in all the functional land uses. The green spaces in low-density residential areas were more complex and irregular in shape compared to other functional land use categories. In comparison, the lowest green space shape complexities were observed in the quasi-residential areas. The complexity of green spaces in low-density residential areas may be attributed to the abundance of greenery in these areas, hence providing a variety of shapes. Studies have shown that more complex and irregularly shaped green spaces may provide higher aesthetic, recreational, ecological and health benefits since they interact well with the surrounding area [55–57]. This implies that people who live in the low-density residential areas are more advantaged than those in the informal/quasi-residential areas of Lilongwe City.

Additionally, the study revealed that the largest patch index values were comparatively low across all functional land uses. Most of the functional land uses had LPIs of less than 3.0. Again, densely populated areas showed the lowest LPIs, which may be attributed to more structures and people living in these places leading to fragmentation. The low LPI also highlight the possibility of disparities in how different land use classes are supplied with green spaces.

The analysis of the mean Euclidean nearest neighbour distance (ENN_MN) indicated that quasi-residential areas had the highest value of 45.54; similarly, quasi-residential and high-rise flat residential areas are more dispersed or farther apart than other land uses. A high value of ENN_MN indicates that green areas are sparsely clustered or concentrated within quasi-residential land use zones. On the other hand, low-density residential areas had the lowest ENN_MN values.

The composition and configuration of urban green spaces in functional land uses highlight the challenges and underscore the need for sustainable urban planning and development to ensure equity in the distribution and accessibility of UGS to all city dwellers. Urban dwellers who are deprived of the many benefits of UGS may be concerned about uneven distribution and a lack of UGS. Furthermore, insufficient green spaces in densely populated areas can worsen already-existing health and environmental injustice disparities.

## The association between urban green space and urban form metrics and implications for urban planning

The study revealed significant correlations between urban green space and urban form. For instance, BD is strongly positively correlated with NP, SHAPE_MN, and ENN_MN. This suggests that as building density increases, the number of patches (NP) of green spaces tends to increase as well. In addition, in areas with higher building density, the green spaces are farther apart from each other, with their shapes becoming more complex. The severity of fragmentation increases with the number of green space patches [58]. The high dispersion and isolation of UGS patches may be associated with the increase in building density. This finding agrees with the finding of Yeh et al. [59], who found that the distances between green spaces in highly urbanised areas were higher than those in less urbanised areas in the Taipei Metropolitan, Taiwan.

BC was strongly positively correlated with LPI. This implies that as the areas within the city become more urbanised and built-up (increased building coverage), larger and more connected green spaces may be present. Similarly, the building perimeter area ratio (MN_PARA), which is a measure of the complexity of the built-up area, showed a significant negative relationship with the PD of green space. We found that areas with compact building forms were less fragmented. This could be due to urban planning interventions that prioritise larger green areas within the central areas of the city [27]. Lilongwe City was built on the garden city concept and has abundant large patches of green space, such as nature sanctuaries, botanic gardens, and golf courses within the central part of the city [27]. In contrast to the findings of Huang et al. [45] and Bereitschaft & Debbage [40], who found that cities with complex urban areas (higher PARA values) had highly fragmented UGS. Compact building forms such as high-rise buildings and mixed-use developments may improve UGS efficiency since they use land more effectively and free up more space for green spaces [60]. This is accomplished by checking urban sprawl and maximising land usage, which increases the overall green space per capita in denser neighbourhoods. However, compact building forms may result in higher population densities, which may lead to the loss of existing greenery to pave the way for buildings and infrastructure to accommodate large populations. Compact building forms can cause UGS fragmentation.

The relationship between the distribution and availability of UGS and urban form and design can be bidirectional. Urban green space is significantly correlated with urban form, which suggests that urban form is crucial to understanding the distribution and configuration of UGS. Urban development plans should incorporate green spaces and consider their significance. Smart growth strategies that prioritise compact and mixed-use development can help with this [60,61]. These strategies seek to preserve or improve green spaces while increasing the density of roads, and both density and coverage of buildings. Integrating UGS into urban fabric improves cities' aesthetic appeal and supports biodiversity and ecological conservation [62]. The purpose of the present research was to establish relationships between UGS and urban form. Future research should consider establishing a cause-and-effect relationship between urban form and UGS.

Our study not only highlights the challenges in the provisioning of UGS in this specific urban context of Lilongwe City, but it also echoes broader global trends in environmental justice and urban planning studies that have developed over time. By cross-referencing our findings with those from other regions, we contribute to a more robust validation of core principles of urban planning, particularly the need for equitable access to green spaces. This cross-contextual validation demonstrates the universality of challenges such as inequitable green space distribution and the impact of urban form on the quality and accessibility of UGS.

## Conclusions

This study revealed that urban green spaces are distributed unevenly among different neighbourhoods and functional land uses in Lilongwe City, despite the city's abundance of green spaces. A sizeable fraction of the populace resides in places with inadequate access to the recommended amounts of green space per person. The study also showed that population density is one of the factors influencing the availability and distribution of UGS as low green space coverage was observed in highly populated neighbourhoods. A low supply of UGS in low-income, high-density and informal settlements may raise concerns about environmental justice. Therefore, deliberate efforts should be undertaken to establish green areas in neighbourhoods with lower urban green space coverage to enhance the liveability and inequities in the distribution of UGS in Lilongwe City. The study has also shown that the spatial composition

and configuration of UGSs are complex and differ with land use class. However, low-density and quasi-residential land uses were found to have more complex green spaces. It was also revealed that urban form had an impact on the composition and configuration of UGS. Urban form indicators such as building coverage, building density, and road density had significant associations with the distribution and configuration of urban green spaces. The study integrated urban form metrics with UGS metrics, a topic that is less explored in the context of developing cities like Lilongwe thereby bringing a novel perspective to the literature on urban planning. Urban form and green space have a complicated and multifaceted interaction. Therefore, urban planning and designs that consider the built environment as well as green spaces as essential elements are necessary to achieve sustainability and liveability in the city. To ensure that there is equitable distribution and accessibility, green spaces should be integrated into the urban fabric. This can be achieved by promoting urban smart growth strategies and policies, which can help strike a balance between development and the preservation of open space while promoting compact and mixed-use development.

## Author Contributions

**Conceptualization:** Odala Nambazo, Kennedy Nazombe.

**Data curation:** Odala Nambazo.

**Formal analysis:** Odala Nambazo.

**Investigation:** Odala Nambazo.

**Methodology:** Odala Nambazo, Kennedy Nazombe.

**Resources:** Odala Nambazo.

**Software:** Odala Nambazo.

**Validation:** Odala Nambazo.

**Visualization:** Odala Nambazo.

**Writing – original draft:** Odala Nambazo.

**Writing – review & editing:** Odala Nambazo, Kennedy Nazombe.

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
