## [Decision Letter · Decision Letter 0]

26 Feb 2024

PONE-D-23-36331The spatial heterogeneity of urban green space distribution and configuration in Lilongwe City, MalawiPLOS ONE

Dear Dr. Nambazo,

Thank you for submitting your manuscript to PLOS ONE. After careful consideration, we feel that it has merit but does not fully meet PLOS ONE’s publication criteria as it currently stands. Therefore, we invite you to submit a revised version of the manuscript that addresses the points raised during the review process.

Reviewer #1: It seems that the study is structured and coherent, and the process from question to answer has been logically progressed in the direction of the research. The only problem is the lack of tables that show some of the analysis and results more clearly, and it is recommended to fix this deficiency in the comments of the attached file.

Reviewer #2: This work started with an interesting concept but was not very well executed or presented. I would like to kindly ask the authors to make further efforts to revise their manuscript based on the following comments:

Notwithstanding the manuscript's endeavor to delineate functional land use categories, it either parallels pre-existing literature or falls short of introducing profound novel insights. Scholarly manuscripts are esteemed for their novelty and seminal contributions. Redundancies with extant literature or the absence of innovative perspectives diminish its perceived value.

The field of management of green spaces offers a variety of methodologies and strategies. Attempting to encapsulate or categorize these within a single manuscript is commendably ambitious but poses a quandary—the balance between comprehensiveness and depth. While the manuscript aims to provide a comprehensive perspective, it risks compromising depth, critical assessment, or intricate explication of specific methodologies. The hallmark of academic literature lies in its depth, rigorous analysis, and groundbreaking revelations. Overextension risks diluting these attributes, offering only superficial overviews without profound insights into specific subjects or methodologies.

The findings remain mostly generic and descriptive. Some interesting considerations are reported in the concluding section, but these are not adequately articulated and backed by data.

We look forward to receiving your revised manuscript.

Kind regards,

Saeid Norouzian-Maleki, Ph.D.

Academic Editor

PLOS ONE

4. Please include the reference section of your manuscript.

5. We note that Figure(s) 1, 2 and 3 in your submission contain [map/satellite] images which may be copyrighted. All PLOS content is published under the Creative Commons Attribution License (CC BY 4.0), which means that the manuscript, images, and Supporting Information files will be freely available online, and any third party is permitted to access, download, copy, distribute, and use these materials in any way, even commercially, with proper attribution. For these reasons, we cannot publish previously copyrighted maps or satellite images created using proprietary data, such as Google software (Google Maps, Street View, and Earth). For more information, see our copyright guidelines: http://journals.plos.org/plosone/s/licenses-and-copyright.

a. You may seek permission from the original copyright holder of Figure(s) 1, 2 and 3 to publish the content specifically under the CC BY 4.0 license.  

Reviewers' comments:

Reviewer's Responses to Questions

**Comments to the Author**

1. Is the manuscript technically sound, and do the data support the conclusions?

Reviewer #1: Yes

Reviewer #2: Partly

2. Has the statistical analysis been performed appropriately and rigorously? 

Reviewer #1: Yes

Reviewer #2: Yes

3. Have the authors made all data underlying the findings in their manuscript fully available?

Reviewer #1: No

Reviewer #2: No

4. Is the manuscript presented in an intelligible fashion and written in standard English?

Reviewer #1: Yes

Reviewer #2: No

5. Review Comments to the Author

Reviewer #1: It seems that the study is structured and coherent, and the process from question to answer has been logically progressed in the direction of the research. The only problem is the lack of tables that show some of the analysis and results more clearly, and it is recommended to fix this deficiency in the comments of the attached file.

Reviewer #2: This work started with an interesting concept but was not very well executed or presented. I would like to kindly ask the authors to make further efforts to revise their manuscript based on the following comments:

Notwithstanding the manuscript's endeavor to delineate functional land use categories, it either parallels pre-existing literature or falls short of introducing profound novel insights. Scholarly manuscripts are esteemed for their novelty and seminal contributions. Redundancies with extant literature or the absence of innovative perspectives diminish its perceived value.

The field of management of green spaces offers a variety of methodologies and strategies. Attempting to encapsulate or categorize these within a single manuscript is commendably ambitious but poses a quandary—the balance between comprehensiveness and depth. While the manuscript aims to provide a comprehensive perspective, it risks compromising depth, critical assessment, or intricate explication of specific methodologies. The hallmark of academic literature lies in its depth, rigorous analysis, and groundbreaking revelations. Overextension risks diluting these attributes, offering only superficial overviews without profound insights into specific subjects or methodologies.

The findings remain mostly generic and descriptive. Some interesting considerations are reported in the concluding section, but these are not adequately articulated and backed by data.

6. PLOS authors have the option to publish the peer review history of their article (what does this mean?). If published, this will include your full peer review and any attached files.

Reviewer #1: No

Reviewer #2: No

---

## [Author Response · Author response to Decision Letter 0]

1 Jun 2024

Response: We have carefully checked the PLOS ONE style templates and we have adjusted throughout the manuscript to fulfill the requirements.

Response: We believe this is not applicable in our case since we did not do any coding.

Response: We have created an account and uploaded our datasets to Mendeley Data. This is an open-access repository. The data can be accessed through the following doi: 10.17632/rfkv3y4237.1

4. Please include the reference section of your manuscript.

Response: A reference section has been added.

5. We note that Figure(s) 1, 2 and 3 in your submission contain [map/satellite] images which may be copyrighted. All PLOS content is published under the Creative Commons Attribution License (CC BY 4.0), which means that the manuscript, images, and Supporting Information files will be freely available online, and any third party is permitted to access, download, copy, distribute, and use these materials in any way, even commercially, with proper attribution. For these reasons, we cannot publish previously copyrighted maps or satellite images created using proprietary data, such as Google software (Google Maps, Street View, and Earth). For more information, see our copyright guidelines: http://journals.plos.org/plosone/s/licenses-and-copyright.

a. You may seek permission from the original copyright holder of Figure(s) 1, 2 and 3 to publish the content specifically under the CC BY 4.0 license. 

Response. In our revised manuscript we added a different Figure (Fig 1) on page 5, which shows the location of the study area. The data used to prepare the maps is publicly available. We properly referred to the data sources of the map within the map’s labels. Figures 2 and 3 have been removed. 

Response: All necessary information has been included within the main manuscript.

Reviewer's Responses to Questions

Comments to the Author

1. Is the manuscript technically sound, and do the data support the conclusions?

Reviewer #1: Yes

Reviewer #2: Partly

Response: We have included a methodological framework of the manuscript to better connect all the elements of our study on page 5 as Figure 2. We have also included Table 6 on page 12 showing detailed statistical results that supports further our observed results with our conclusions.

2. Has the statistical analysis been performed appropriately and rigorously?

Reviewer #1: Yes

Reviewer #2: Yes

3. Have the authors made all data underlying the findings in their manuscript fully available?

Reviewer #1: No

Reviewer #2: No

Response: As already stated in the response to the Academic Editor, we have made all underlining data fully available in the Mendeley Data open-access repository. The following is the doi: 10.17632/rfkv3y4237.1

4. Is the manuscript presented in an intelligible fashion and written in standard English?

Reviewer #1: Yes

Reviewer #2: No

Response: The English grammar of the revised version has been greatly improved.

5. Review Comments to the Author

Reviewer #1: It seems that the study is structured and coherent, and the process from question to answer has been logically progressed in the direction of the research. The only problem is the lack of tables that show some of the analysis and results more clearly, and it is recommended to fix this deficiency in the comments of the attached file.

Response: Thank you for your constructive feedback. We introduced Table 3 on page 8 to provide an overview of the urban form metrics used. To further elucidate the relationship between population density and green space configuration, we added Table 6 on page 12. A diagram of the process has also been included on page 5. 

Reviewer #2: This work started with an interesting concept but was not very well executed or presented. I would like to kindly ask the authors to make further efforts to revise their manuscript based on the following comments:

Notwithstanding the manuscript's endeavor to delineate functional land use categories, it either parallels pre-existing literature or falls short of introducing profound novel insights. Scholarly manuscripts are esteemed for their novelty and seminal contributions. Redundancies with extant literature or the absence of innovative perspectives diminish its perceived value.

The field of management of green spaces offers a variety of methodologies and strategies. Attempting to encapsulate or categorize these within a single manuscript is commendably ambitious but poses a quandary—the balance between comprehensiveness and depth. While the manuscript aims to provide a comprehensive perspective, it risks compromising depth, critical assessment, or intricate explication of specific methodologies. The hallmark of academic literature lies in its depth, rigorous analysis, and groundbreaking revelations. Overextension risks diluting these attributes, offering only superficial overviews without profound insights into specific subjects or methodologies.

The findings remain mostly generic and descriptive. Some interesting considerations are reported in the concluding section, but these are not adequately articulated and backed by data.

Response: We appreciate the insightful comments and constructive critiques you provided. We recognize the importance of emphasising the unique contributions of our study, especially in a well-researched field. To address this concern, we have revised the introduction and discussion sections to better highlight the unique aspects of our research. We clarified that our study is one of the first to comprehensively analyse urban green space distribution in Lilongwe, providing new data and insights from a lesser-studied geographical context where UGS research is scarce (page 3 line 68). We expanded on how our work integrates urban form metrics with UGS analysis, which is less explored in the context of developing cities like Lilongwe, adding a novel perspective to urban planning literature (page 20 line 447). In addition, we emphasised our findings on the inequities in green space distribution and how they contribute to discussions on environmental justice in urban settings, particularly in developing countries like Malawi (Page 26 Line 326). Further, by cross-referencing our findings with those from other regions, we contribute to a more robust validation of core principles of urban planning, particularly the need for equitable access to green spaces. This cross-contextual validation demonstrates the universality of challenges such as inequitable green space distribution and the impact of urban form on the quality and accessibility of UGSs (Page 20 Line 425 and throughout the discussion). We have added a new table to better support our claims and provide clearer, more detailed insights into the relationship between population and urban greenspaces in Lilongwe (Table 6 on page 12).

---

## [Decision Letter · Decision Letter 1]

17 Jun 2024

PONE-D-23-36331R1The spatial heterogeneity of urban green space distribution and configuration in Lilongwe City, MalawiPLOS ONE

Dear Dr. Nambazo,

Thank you for submitting your manuscript to PLOS ONE. After careful consideration, we feel that it has merit but does not fully meet PLOS ONE’s publication criteria as it currently stands. Therefore, we invite you to submit a revised version of the manuscript that addresses the points raised during the review process.

Reviewer #1: The added tables and especially the prepared diagram helped me understand the topic better. However, figures 2 and 3 would have been better if they had not been deleted because they better showed the population distribution and green space.

Reviewer #2: The manuscript describe a technically sound piece of scientific research with data that supports the conclusions. All comments have been addressed in the revised manuscript.

We look forward to receiving your revised manuscript.

Kind regards,

Saeid Norouzian-Maleki, Ph.D.

Academic Editor

PLOS ONE

Journal Requirements:

Reviewers' comments:

Reviewer's Responses to Questions

**Comments to the Author**

1. If the authors have adequately addressed your comments raised in a previous round of review and you feel that this manuscript is now acceptable for publication, you may indicate that here to bypass the “Comments to the Author” section, enter your conflict of interest statement in the “Confidential to Editor” section, and submit your "Accept" recommendation.

Reviewer #1: (No Response)

Reviewer #2: All comments have been addressed

2. Is the manuscript technically sound, and do the data support the conclusions?

Reviewer #1: Yes

Reviewer #2: Yes

3. Has the statistical analysis been performed appropriately and rigorously? 

Reviewer #1: Yes

Reviewer #2: Yes

4. Have the authors made all data underlying the findings in their manuscript fully available?

Reviewer #1: Yes

Reviewer #2: Yes

5. Is the manuscript presented in an intelligible fashion and written in standard English?

Reviewer #1: Yes

Reviewer #2: Yes

6. Review Comments to the Author

Reviewer #1: The added tables and especially the prepared diagram helped me understand the topic better. However, figures 2 and 3 would have been better if they had not been deleted because they better showed the population distribution and green space.

Reviewer #2: The manuscript describe a technically sound piece of scientific research with data that supports the conclusions. All comments have been addressed in the revised manuscript.

7. PLOS authors have the option to publish the peer review history of their article (what does this mean?). If published, this will include your full peer review and any attached files.

Reviewer #1: No

Reviewer #2: No

---

## [Author Response · Author response to Decision Letter 1]

4 Jul 2024

Reviewer #1: The added tables and especially the prepared diagram helped me understand the topic better. However, figures 2 and 3 would have been better if they had not been deleted because they better showed the population distribution and green space.

Response: We have reinstated figures 2 and 3, now figures 3 and 4 respectively. These figures have been prepared by the authors. Figure 3 is derived from sentinel 2 images. These images have no restrictions on reuse, sale, or redistribution. They only require that the author include a statement of the data source in their manuscripts. Figure 4 was prepared from the public report, the Malawi population and housing census 2018.

---

## [Editor Report · Decision Letter 2]

8 Jul 2024

The spatial heterogeneity of urban green space distribution and configuration in Lilongwe City, Malawi

PONE-D-23-36331R2

Dear Dr. Nambazo,

We’re pleased to inform you that your manuscript has been judged scientifically suitable for publication and will be formally accepted for publication once it meets all outstanding technical requirements.

Kind regards,

Saeid Norouzian-Maleki, Ph.D.

Academic Editor

PLOS ONE
---

## [Editor Report · Acceptance letter]

15 Jul 2024

PONE-D-23-36331R2 

PLOS ONE

Dear Dr. Nambazo, 

I'm pleased to inform you that your manuscript has been deemed suitable for publication in PLOS ONE. Congratulations! Your manuscript is now being handed over to our production team.

Kind regards, 

on behalf of

Dr. Saeid Norouzian-Maleki 

Academic Editor

PLOS ONE